# SELF-ENSEMBLE PROTECTION: TRAINING CHECKPOINTS ARE GOOD DATA PROTECTORS

**Sizhe Chen**[1,2]**, Geng Yuan**[2]**, Xinwen Cheng**[1]**, Yifan Gong**[2]**, Minghai Qin**[2]**, Yanzhi Wang**[2]**, Xiaolin Huang**[1]*

[1]Department of Automation, Shanghai Jiao Tong University
[2]Department of Electrical and Computer Engineering, Northeastern University

## ABSTRACT

As data becomes increasingly vital, a company would be very cautious about releasing data, because the competitors could use it to train high-performance models, thereby posing a tremendous threat to the company's commercial competence. To prevent training good models on the data, we could add imperceptible perturbations to it. Since such perturbations aim at hurting the entire training process, they should reflect the vulnerability of DNN training, rather than that of a single model. Based on this new idea, we seek perturbed examples that are always unrecognized (never correctly classified) in training. In this paper, we uncover them by model checkpoints' gradients, forming the proposed self-ensemble protection (SEP), which is very effective because (1) learning on examples ignored during normal training tends to yield DNNs ignoring normal examples; (2) checkpoints' cross-model gradients are close to orthogonal, meaning that they are as diverse as DNNs with different architectures. That is, our amazing performance of ensemble only requires the computation of training one model. By extensive experiments with 9 baselines on 3 datasets and 5 architectures, SEP is verified to be a new state-of-the-art, e.g., our small $\ell_\infty = 2/255$ perturbations reduce the accuracy of a CIFAR-10 ResNet18 from 94.56% to 14.68%, compared to 41.35% by the best-known method. Code is available at `https://github.com/Sizhe-Chen/SEP`.

## 1 INTRODUCTION

Large-scale datasets have become increasingly important in training high-performance deep neural networks (DNNs). Thus, it is a common practice to collect data online (Mahajan et al., 2018; Sun et al., 2017), an almost unlimited data source. This poses a great threat to the commercial competence of data owners such as social media companies since the competitors could also train good DNNs from their data. Therefore, great efforts have been devoted to protecting data from unauthorized use in model training. The most typical way is to add imperceptible perturbations to the data, so that DNNs trained on it have poor generalization (Huang et al., 2020a; Fowl et al., 2021b).

Existing data protection methods use a single DNN to generate incorrect but DNN-sensitive features (Huang et al., 2020a; Fu et al., 2021; Fowl et al., 2021b) for training data by, *e.g.*, adversarial attacks (Goodfellow et al., 2015). However, the data protectors cannot know what DNN and what training strategies the unauthorized users will adopt. Thus, the protective examples should aim at **hurting the DNN training, a whole dynamic process, instead of a static DNN**. Therefore, it would be interesting to study the *vulnerability of DNN training*. Recall that the vulnerability of a DNN is revealed by the adversarial examples which are similar to clean ones but unrecognized by the model (Madry et al., 2018). Similarly, we depict the vulnerability of training by the perturbed training samples that are never predicted correctly during training. Learning on examples ignored during normal training tends to yield DNNs ignoring normal examples.

Such examples could be easily uncovered by the gradients from the ensemble of model training checkpoints. However, ensemble methods have never been explored in data protection to the best of our knowledge, so it is natural to wonder

*Can we use these intermediate checkpoint models for data protection in a **self-ensemble** manner?*

---

*Correspondence to Xiaolin Huang (xiaolinhuang@sjtu.edu.cn).

An effective ensemble demands high diversity of sub-models, which is generally quantified by their gradient similarity (Pang et al., 2019; Yang et al., 2021), *i.e.*, the gradients on the same image from different sub-models should be orthogonal. Surprisingly, we found that checkpoints' gradients are as orthogonal as DNNs with different architectures in the conventional ensemble. In this regard, we argue that intermediate checkpoints are very diverse to form the proposed self-ensemble protection (SEP), challenging existing beliefs on their similarity (Li et al., 2022).

By SEP, effective ensemble protection is achieved by the computation of training only one DNN. Since the scale of data worth protecting is mostly very large, SEP avoids tremendous costs by training multiple models. Therefore, our study enables a practical ensemble for large-scale data, which may help improve the generalization, increase the attack transferability, and study DNN training dynamics.

Multiple checkpoints offer us a pool of good features for an input. Thus, we could additionally take the advantage of diverse features besides diverse gradients at no cost. Inspired by neural collapse theory (Papyan et al., 2020), which demonstrates that the mean feature of samples in a class is a highly representative depiction of this class, we bring about a novel feature alignment loss that induces a sample's last-layer feature collapse into the mean of incorrect-class features. With features from multiple checkpoints, FA robustly injects incorrect features so that DNNs are deeply confounded.

Equipping SEP with FA, our method achieves astonishing performance by revealing the vulnerability of DNN training: (1) our examples are mostly mis-classified in any training processes compared to a recent method (Sandoval-Segura et al., 2022), and (2) clean samples are always much closer to each other than to protected samples, indicating that the latter belong to another distribution that could not be noticed by normal training. By setting $\ell_\infty = 2/255$, a very small bound, SEP perturbations on the CIFAR-10 training set reduce the testing accuracy of a ResNet18 from 94.56% to 14.68%, while the best-known results could only reach 41.35% with the same amount of overall calculation to craft the perturbations. The superiority of our method is also observable in the study on CIFAR-100 and ImageNet subset on 5 architectures. We also study perturbations under different norms, and found that mixing $\ell_\infty$ and $\ell_0$ perturbations (Wu et al., 2023) is the only effective way to resist $\ell_\infty$ adversarial training, which could recover the accuracy for all other types of perturbations. Our contributions could be summarized below.

- We propose that protective perturbations should reveal the vulnerability of the DNN training process, which we depict by the examples never classified correctly in training.

- We uncover such examples by the self-ensemble of model checkpoints, which are found to be surprisingly diverse as data protectors.

- Our method is very effective even using the computation of training one DNN. Equipped with a novel feature alignment loss, our $\ell_\infty = 8/255$ perturbations lead DNNs to have < 5.7% / 3.2% / 0.6% accuracy on CIFAR-10 / CIFAR-100 / ImageNet subset.

## 2 RELATED WORK

Small perturbations are known to be able to fool DNNs into incorrect predictions (Szegedy et al., 2014). Such test-time adversarial perturbations are crafted effectively by adversarially updating samples with model gradients (Carlini & Wagner, 2017), and the produced adversarial examples (AEs) transfer to hurt other DNNs as well (Chen et al., 2022). Similarly, training-time adversarial perturbations, *i.e.*, poisoning examples, are also obtainable by adversarially modify training samples using DNN gradients (Koh & Liang, 2017; Fowl et al., 2021b). All DNNs trained on poisoning examples generalize poorly on clean examples, making poisoning methods helpful in protecting data from unauthorized use of training. Besides adversarial noise, it has been demonstrated that error-minimization (Huang et al., 2020a), gradient alignment (Fowl et al., 2021a) and influence functions (Fang et al., 2020) are also useful in protecting data. However, current methods only use one DNN because the scale of data worth protection is very large for training multiple models.

Ensemble is validated as a panacea for boosting adversarial attacks (Liu et al., 2017; Dong et al., 2018). By aggregating the probabilities (Liu et al., 2017), logits or losses (Dong et al., 2018) of multiple models, ensemble attacks significantly increase the black-box attack success rate. Ensemble attacks could be further enhanced by reducing the gradient variance of sub-models (Xiong et al., 2022), and such an optimization way is also adopted in our method. Besides, ensemble has also

been shown effective as a defense method by inducing low diversity across sub-models (Pang et al., 2019; Yang et al., 2020; 2021) or producing diverse AEs in adversarial training (Tramèr et al., 2018; Wang & Wang, 2021). Despite the good performance of ensemble in attacks and defenses, it has not been introduced to protect datasets due to its inefficiency. In this regard, we adopt the self-ensemble strategy, which only requires the computation of training one DNN. Its current applications are focused on semi-supervised learning (Zhao et al., 2019; Liu et al., 2022).

Two similar but different tasks besides poisoning-based data protection are adversarial training and backdoor attacks. Adversarial training (Madry et al., 2018; Zhang et al., 2019; Stutz et al., 2020) continuously generates AEs with current checkpoint gradients to improve the model's robustness towards worst-case perturbations. In contrast, data protection produces fixed poisoning examples so that unauthorized training yields low clean accuracy. Backdoor attacks (Geiping et al., 2020; Huang et al., 2020b) perturb a small proportion of the training set to make the DNNs mispredict certain samples, but remain well-functional on other clean samples. While data protectors perturb the whole training set to degrade the model's performance on all clean samples.

## 3 THE PROPOSED METHOD

### 3.1 PRELIMINARIES

We first introduce the threat model and problem formulation of the data protection task. The data owner company wishes to release data for users while preventing an unauthorized *appropriator* to collect data for training DNNs. Thus, the data *protector* would add imperceptible perturbations to samples, so that humans have no obstacle to seeing the data, while the appropriator cannot train DNNs to achieve an acceptable testing accuracy. Since the protector has access to the whole training set, it can craft perturbations for each sample for effective protection (Shen et al., 2019; Feng et al., 2019; Huang et al., 2020a; Fowl et al., 2021b). Mathematically, the problem can be formulated as

$$\max_{\boldsymbol{\delta} \in \Pi_\varepsilon} \sum_{(\boldsymbol{x},y) \in \mathcal{D}} \mathcal{L}_{\mathrm{CE}}(f_\mathrm{a}(\boldsymbol{x}, \theta^*), y), \quad \text{s.t. } \theta^* \in \arg\min_\theta \sum_{(\boldsymbol{x},y) \in \mathcal{T}} \mathcal{L}_{\mathrm{CE}}\left(f_\mathrm{a}\left(\boldsymbol{x} + \boldsymbol{\delta}, \theta\right), y\right), \tag{1}$$

where the perturbations $\boldsymbol{\delta}$ bounded by $\varepsilon$ are added to the training set $\mathcal{T}$ so that an appropriator model trained on it $f_\mathrm{a}(\cdot, \theta^*)$ have a low accuracy on the test set $\mathcal{D}$, *i.e.*, a high cross-entropy loss $\mathcal{L}_{\mathrm{CE}}(\cdot, \cdot)$. The $\boldsymbol{\delta}$ could be effectively calculated by targeted attacks (Fowl et al., 2021b), which use a well-trained protecting DNN $f_\mathrm{p}$ to produce targeted adversarial examples (AEs) that have the non-robust features in the incorrect class $g(y)$ as

$$\boldsymbol{x}_{t+1} = \Pi_\epsilon\left(\boldsymbol{x}_t - \alpha \cdot \mathrm{sign}\left(\boldsymbol{G}_{\mathrm{CE}}(f_\mathrm{p}, \boldsymbol{x}_t)\right)\right), \quad \boldsymbol{G}_{\mathrm{CE}}(f_\mathrm{p}, \boldsymbol{x}) = \nabla_{\boldsymbol{x}} \mathcal{L}_{\mathrm{CE}}\left(f_\mathrm{p}\left(\boldsymbol{x}\right), g(y)\right), \tag{2}$$

where $\Pi_\epsilon$ clips the sample into the $\varepsilon$ $\ell_\infty$-norm bound after each update with a step size $\alpha$. $g(\cdot)$ stands for a permutation on the label space. Here our method also adopts the optimization in (2).

### 3.2 DEPICTING THE VULNERABILITY OF DNN TRAINING

Current methods (Shen et al., 2019; Feng et al., 2019; Huang et al., 2020a; Fowl et al., 2021b) craft protective perturbations that are supposed to generalize to poison different unknown architectures by a single DNN $f_\mathrm{p}$. However, the data protectors cannot know what DNN and what training strategies the unauthorized users will adopt. Thus, the protective examples should aim at hurting the DNN training, a whole dynamic process, instead of a static DNN. Therefore, it would be interesting to study the vulnerability of DNN training.

Recall that the vulnerability of a DNN is represented by AEs (Madry et al., 2018), because they are slightly different from clean testing samples but are totally unrecognizable by a static model. Similarly, the vulnerability of DNN training could be depicted by examples that are slightly different from clean training samples but are always unrecognized in the training process, *i.e.*, the perturbed data never correctly predicted by the training model. If we view the training process as the generation of checkpoint models, the problem becomes finding the examples that are adversarial to checkpoints, which could be easily solved by the ensemble attack (Dong et al., 2018).

Let us investigate whether the training checkpoints, which are similar (Li et al., 2022) in architecture and parameters, could be diverse sub-models for effective self-ensemble. To measure the diversity,

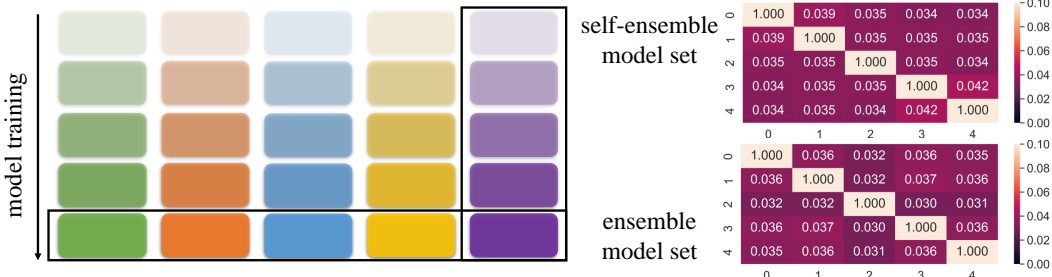

Figure 1: Ensemble and Self-Ensemble. Ensemble trains multiple models (left to right and top to bottom), while self-ensemble trains once and collects intermediate checkpoints (only top to bottom). Thus, self-ensemble costs much less training calculation. Moreover, checkpoints could provide diverse gradients. The right figures show the average absolute cosine value of gradients (on 1K CIFAR-10 images) for each model pair in ensemble (bottom) and self-ensemble (top) sub-models. In ensemble, models 0 to 4 stand for ResNet18, SENet18, VGG16, DenseNet121, and GoogLeNet, respectively, and in self-ensemble, they mean the ResNet18 after training for 24, 48, 72, 96, 120 epochs. Gradient analysis on CIFAR-100 and ImageNet are in Appendix B.

we adopt the common gradient similarity metric (Pang et al., 2019; Yang et al., 2021). In Fig. 1 (upper right), we plot the average of absolute cosine value for gradients in different checkpoints, and the low value indicates that gradients on images are close to orthogonal for different checkpoints like in the ensemble of different architectures (bottom right). This means, surprisingly, intermediate checkpoints are very diverse and sufficient to form the proposed self-ensemble protection (SEP) as

$$\boldsymbol{G}_{\text{SEP}}(f_{\text{p}}, \boldsymbol{x}) = \sum_{k=0}^{n-1} \boldsymbol{G}_{\text{CE}}(f_{\text{p}}^{k}, \boldsymbol{x}), \tag{3}$$

where $f_{\text{p}}^{k}$ is the $k^{\text{th}}$ equidistant intermediate checkpoint and $\boldsymbol{G}_{\text{SEP}}$ is the gradients for update in (2). As illustrated in 1 (left), SEP (vertical box) requires the computation of training only one DNN compared to the conventional ensemble (horizontal box) that needs time- and resource-consuming training processes to obtain a large number of ensemble models. This efficiency is especially important for data protection. Because only a large amount of data, if stolen, could be used to train competitive DNNs. In this regard, the scale of data requiring particular protection would be large, and saving the calculation of training extra models makes a significant difference.

SEP is differently motivated compared to conventional ensemble. Ensemble attacks aim to produce architecture-invariant adversarial examples, and such transferable examples reveal the common vulnerability of different architectures (Chen et al., 2022). SEP, in contrast, targets the vulnerability of DNN training. By enforcing consistent misclassification, SEP produces examples ignored during normal training, and learning on them would thus yield DNNs ignoring normal examples.

### 3.3 PROTECTING DATA BY SELF-ENSEMBLE

Multiple checkpoints offer us a pool of features for an input. Those representations, though distinctive, all contribute to accurate classification. Thus, we could additionally take the advantage of diverse features besides diverse gradients at no cost. Motivated by this, we resort to the neural collapse theory (Papyan et al., 2020) because it unravels the characteristics of DNN features.

Neural collapse has four manifestations for a deep classifier. (1) In-class variability of last-layer activation collapse to class means. (2) Class means converge to simplex equiangular tight frame. (3) Linear classifiers approach class means. (4) Classifier converges to choose the nearest class mean. They demonstrate that the last-layer features of well-trained DNNs center closely on class means. In this regard, the mean feature of in-class samples is a highly representative depiction of this class.

Based on this, we develop the feature alignment loss to jointly use different but good representations of a class from multiple checkpoints. Specifically, for every checkpoint, we encourage the last-layer feature of a sample to approximate the mean feature of target-class samples. In this way, FA promotes neural collapse to incorrect centers so that a sample has the exact high-dimensional feature of the

target-class samples. Therefore, non-robust features of that target class could be robustly injected into data so that DNNs are deeply confounded. Mathematically, FA in SEP can be expressed as

$$\boldsymbol{G}_{\text{SEP-FA}}(h_{\text{p}}, \boldsymbol{x}) = \sum_{k=0}^{n-1} \boldsymbol{G}_{\text{FA}}(h_{\text{p}}^k, \boldsymbol{x}) = \sum_{k=0}^{n-1} \nabla_{\boldsymbol{x}} \|h_{\text{p}}^k(\boldsymbol{x}) - h_{\text{c}}^k(g(y))\|, \quad h_{\text{c}}^k(y) = \frac{\sum_{\boldsymbol{x} \in \mathcal{T}_y} h_{\text{p}}^k(\boldsymbol{x})}{|\mathcal{T}_y|}, \quad (4)$$

where $h_{\text{p}}^k$ stands for the feature extractor (except for the last linear layer) of $f_{\text{p}}^k$, and $h_{\text{c}}^k(g(y))$ is for the mean (center) feature in the target class $g(y)$ calculated by $h_{\text{p}}^k$. $\|\cdot\|$ means the MSE loss.

Our overall algorithm is summarized in Alg. 1, where we use a stochastic variance reduction (VR) gradient method (Johnson & Zhang, 2013) to avoid bad local minima in optimization. Our method first calculates the FA gradients $\boldsymbol{g}_k$ by each training checkpoint (line 4). Then before updating the sample by the accumulated gradients (line 11), we reduce the variance of ensemble gradients (from $\boldsymbol{g}_{\text{ens}}$ to $\boldsymbol{g}_{\text{upd}}$) in a predictive way by $M$ inner virtual optimizations on $\hat{\boldsymbol{x}}_m$, which has been verified to boost ensemble adversarial attacks (Xiong et al., 2022).

---

**Algorithm 1** Self-Ensemble Protection with Feature Alignment and Variance Reduction

**Input:** Dataset $\mathcal{T} = \{(\boldsymbol{x}, y)\}$, $\ell_\infty$ bound $\varepsilon$, step size $\alpha$, number of protection iterations $T$, number of training iterations $N$, number of checkpoints in self-ensemble $n$, number of inner updates $M$

**Output:** Protected dataset $\boldsymbol{x}'$

1: Train a DNN for $N$ epochs and save $n$ equidistant checkpoints
2: $\boldsymbol{x}_0 = \boldsymbol{x}$
3: **for** $t = 0 \to T - 1$ **do**
4:     **for** $k = 0 \to n - 1$ **do** $\boldsymbol{g}_k = \boldsymbol{G}_{\text{FA}}(h_{\text{p}}^k, \boldsymbol{x}_t)$    *# get the gradients by each checkpoints as (4)*
5:     $\boldsymbol{g}_{\text{ens}} = \frac{1}{n} \sum_{k=0}^{n-1} \boldsymbol{g}_k$, $\boldsymbol{g}_{\text{upd}} = 0$, $\hat{\boldsymbol{x}}_0 = \boldsymbol{x}_t$    *# initialize variables for inner optimization*
6:     **for** $m = 0 \to M - 1$ **do**
7:         Pick a random index $k$    *# stochastic variance reduction (Johnson & Zhang, 2013)*
8:         $\boldsymbol{g}_{\text{upd}} = \boldsymbol{g}_{\text{upd}} + \boldsymbol{G}_{\text{FA}}(h_{\text{p}}^k, \hat{\boldsymbol{x}}_m) - (\boldsymbol{g}_k - \boldsymbol{g}_{\text{ens}})$    *# accumulate gradients with variance*
9:         $\hat{\boldsymbol{x}}_{m+1} = \Pi_\epsilon \left( \hat{\boldsymbol{x}}_m - \alpha \cdot \text{sign}(\boldsymbol{g}_{\text{upd}}) \right)$    *# virtual update on$\hat{\boldsymbol{x}}_m$*
10:     **end for**
11:     $\boldsymbol{x}_{t+1} = \Pi_\epsilon \left( \boldsymbol{x}_t - \alpha \cdot \text{sign}(\boldsymbol{g}_{\text{upd}}) \right)$    *# update samples with variance-reduced gradients*
12: **end for**
13: **return** $\boldsymbol{x}' = \boldsymbol{x}_{T-1}$

---

In a word, the main part of our method is to use checkpoints to craft targeted AEs for the training set (Line 4-5 in Alg. 1) in a self-ensemble protection (SEP) manner. And SEP could be boosted by FA loss (Line 4) and VR optimization (Line 6-11). In this way, our overall method only requires $1 \times N$ training epochs, and $T \times (n + M)$ times of backward calculation to update samples.

## 4 EXPERIMENTS

### 4.1 SETUP

We evaluate SEP along with 7 data protection baselines, including adding random noise, TensorClog aiming to cause gradient vanishing (Shen et al., 2019), Gradient Alignment to target-class gradients (Fowl et al., 2021a), DeepConfuse that protects by an autoencoder (Feng et al., 2019), Unlearnable Examples (ULEs) using error-minimization noise (Huang et al., 2020a), Robust ULEs (RULEs) that use adversarial training (Fu et al., 2021), Adversarial Poison (AdvPoison) resorting to targeted attacks (Fowl et al., 2021b), and AutoRegressive (AR) Poison (Sandoval-Segura et al., 2022) using Markov chain. Hyperparameters of baselines are shown in Appendix D. We use the reproduced results in (Fowl et al., 2021b) in Table 1, 5, and 6.

For our method, we optimize class-$y$ samples to have the mean feature of target incorrect class $g(y)$, where $g(y) = (y + 5)\%10$ for CIFAR-10 (Krizhevsky et al., 2009) and ImageNet (Krizhevsky et al., 2017) protected classes, and $g(y) = (y + 50)\%100$ for CIFAR-100. For the ImageNet subset, we train $f_{\text{a}}$ and $f_{\text{p}}$ with the first 100 classes in ImageNet-1K, but only protect samples in 10 significantly different classes, which are African chameleon, black grouse, electric ray, hammerhead, hen, house finch, king snake, ostrich, tailed frog, and wolf spider. This establishes the class-wise data protection

setting, and the reported accuracy is calculated on the testing samples in these 10 classes. We train a ResNet18 for $N = 120$ epochs as $f_p$ following (Huang et al., 2020a; Fowl et al., 2021b). 15 equidistant intermediate checkpoints (epoch 8, 16, ..., 120) are adopted with $M = 15, T = 30$ if not otherwise stated. Experiments are conducted on an NVIDIA Tesla A100 GPU but could be run on GPUs with 4GB+ memory because we store checkpoints on hardware.

The data protection methods are assessed by training models with 5 architectures, which are ResNet18 (He et al., 2016), SENet18 (Hu et al., 2018), VGG16 (Simonyan & Zisserman, 2015), DenseNet121 (Huang et al., 2017), and GoogLeNet (Szegedy et al., 2015). They are implemented from Pytorch vision (Paszke et al., 2019). We train appropriator DNNs $f_a$ for 120 epochs by an SGD optimizer with an initial learning rate of 0.1, which is divided by 10 in epochs 75 and 90. The momentum item in training is 0.9 and the weight decay is 5e-4. In this setting, DNNs trained on clean data could get great accuracy, *i.e.*, 95% / 75% / 78% for CIFAR-10 / CIFAR-100 / ImageNet subset. Training configurations are the same for $f_a$ and $f_p$. In the ablation study, we denote the pure self-ensemble (3) as SEP, SEP with feature alignment (4) as SEP-FA, and SEP-FA with variance reduction as SEP-FA-VR (Alg. 1). In other experiments, "ours" stands for our final method, *i.e.*, SEP-FA-VR. We put the confusion matrix of $f_a$ in Appendix C to provide class-wise analysis.

## 4.2 Uncovering the vulnerability of DNN training

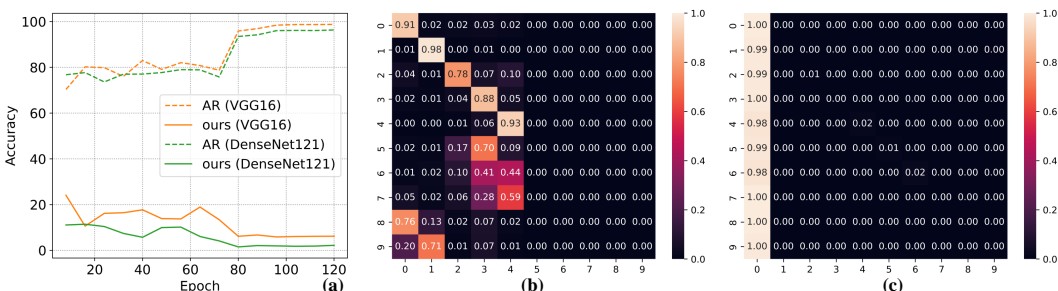

Figure 2: (a) Our examples generated by ResNet18 are mostly mis-classified during the training of DenseNet121 and VGG16 compared to AR poison $\ell_2 = 1$. (b) The confusion matrix (X-axis for predicted classes) of training with clean (classes 0-4) and protective examples (classes 5-9). (c) The confusion matrix of training with clean (class 0) and protective examples (classes 1-9).

We first investigate whether our examples successfully reveal the vulnerability of DNN training. If so, we should be able to hurt different training processes regardless of the model architecture. In this regard, we test the "cross-training transferability" of our protective data. We report the results in Fig. 2 (a), where one could see that SEP samples generated by ResNet18 training tend to be mispredicted in training DenseNet121 and VGG16. In contrast, AR samples behave like clean training data and can be well recognized. This demonstrates that our method well depicts the vulnerability of DNN training compared to the recent baseline (Sandoval-Segura et al., 2022).

We also perform a class-wise study to illustrate how DNNs treat clean and protective examples. We first train a CIFAR-10 DNN with clean samples (classes 0-4) and protective ones (classes 5-9, with features of classes 0-4). The DNN performs well in classes 0-4 but misclassifies all clean samples in classes 5-9 to classes 0-4, seeing Fig. 2 (b). This indicates in DNN's view, clean samples (from different classes) are much closer to each other than to protective ones (which look very similar). More extremely, if we inject features of class 0 to classes 1-9 examples, and use them with clean samples (class 0) to train, the DNN would classify all testing samples to class 0, seeing Fig. 2 (c).

## 4.3 Protective Performance

By depicting the vulnerability of DNN training, our method achieves amazing protective performance on 3 datasets and 5 architectures against various baselines. In table 1, our method surpasses existing state-of-the-art baselines by a large margin, leading DNNs to have < 5.7% / 3.2% / 0.6% accuracy on CIFAR-10 / CIFAR-100 / ImageNet subset. Comparison with weaker baselines is shown in Table 5. The great performance enables us to set an extremely small bound of $\varepsilon = 2/255$, even

for high-resolution ImageNet samples. The perturbations would be invisible even under meticulous inspection, seeing Appendix A. However, the appropriator can only reach no more than 30% accuracy in most cases, seeing Table 2.

Table 1: Model testing accuracy trained on the protected dataset ($\ell_\infty = 8/255$).

| Dataset | CIFAR-10 | | | CIFAR-100 | | | ImageNet subset | | |
|---|---|---|---|---|---|---|---|---|---|
| Model | RN18 | VGG16 | DN121 | RN18 | VGG16 | DN121 | RN18 | VGG16 | DN121 |
| ULEs | 19.93 | 28.34 | 20.25 | 14.81 | 17.56 | 13.71 | 12.20 | 11.14 | 15.44 |
| RULEs | 27.09 | 28.17 | 24.96 | 10.14 | 14.39 | 13.96 | 13.74 | 12.77 | 14.36 |
| AdvPoison | 6.25 | 6.88 | 6.22 | 3.49 | 4.46 | 3.57 | 2.30 | 5.40 | 4.80 |
| Ours | **4.73** | **5.61** | **3.76** | **2.65** | **3.15** | **2.43** | **0.00** | **0.60** | **0.20** |

Table 2: Model testing accuracy trained on slight perturbed dataset ($\ell_\infty = 2/255$).

| Dataset | RN18 | SENet18 | VGG16 | DN121 | GoogLeNet |
|---|---|---|---|---|---|
| CIFAR-10 | 14.68 | 15.93 | 23.66 | 15.02 | 17.99 |
| CIFAR-100 | 21.16 | 19.48 | 66.73 | 21.85 | 27.49 |
| ImageNet subset | 8.80 | 10.70 | 27.00 | 9.40 | 30.80 |

Adversarial training (AT) has been validated as the most effective strategy to recover the accuracy of training with protective perturbations. It does not hinder the practicability of data protection methods because AT significantly decreases the accuracy and requires several-fold training computation. However, it would be interesting to study the effect of different types of perturbations in different AT settings. Here we study with AR Poison, which is claimed to resist AT. We set the perturbation bound as $\ell_2 = 1$ (step size $\alpha = 0.2$) (Sandoval-Segura et al., 2022) and $\ell_0 = 1$ (Wu et al., 2023), and the latter means perturbing one pixel without other restrictions. We keep the AT bound the same as the perturbations bound, and find that in this case, both $\ell_\infty$ and $\ell_2$ ATs could recover accuracy of $\ell_\infty$, $\ell_2$, and $\ell_\infty + \ell_2$ perturbations. The only type of perturbations able to resist $\ell_\infty$ AT is the mixture of $\ell_\infty$ and $\ell_0$ perturbations. Besides, our method is significantly better than AR Poison in normal training.

Table 3: Performance of perturbations under different norms $\ell_\infty = 8/255, \ell_2 = 1, \ell_0 = 1$ in CIFAR-10 adversarial training of ResNet18. $\ell_\infty + \ell_2$ means mixing (adding) perturbations.

| Pert. | $\ell_\infty$ | | | $\ell_2$ | | | $\ell_\infty + \ell_2$ | | | $\ell_\infty + \ell_0$ | | |
|---|---|---|---|---|---|---|---|---|---|---|---|---|
| AT | None | $\ell_\infty$ | $\ell_2$ | None | $\ell_\infty$ | $\ell_2$ | None | $\ell_\infty$ | $\ell_2$ | None | $\ell_\infty$ | $\ell_2$ |
| AR | 20.49 | 84.73 | 82.66 | 12.99 | 82.03 | 82.86 | 12.29 | 82.87 | 83.47 | 15.25 | **12.28** | **70.26** |
| ours | 4.73 | 82.51 | 81.91 | 3.67 | 81.52 | 81.89 | 5.05 | 81.54 | 82.25 | 3.43 | 13.19 | 71.01 |

## 4.4 ABLATION STUDY

We study the performance improvements from SEP, FA, and VR separately along with the best baseline AdvPoison (Fowl et al., 2021b), seeing Table 1. We first control the overall computation the same for all experiments. Then we vary the number of sub-models $n$ to see its effect on our method.

In Table 4, we maintain our methods to have comparative computation with AdvPoison, which trains the protecting DNN for 40 epochs and craft perturbations by 250 steps 8 times (we modify it to 4). In SEP and SEP-FA, we train for $N = 120$ epochs and use $n = 30$ checkpoints to update samples for $T = 30$ times, aligning the computation with AdvPoison. In SEP-FA-VR with inner update $M = 15$, we alter the number of checkpoints $n = 15$ so that the overall computation is the same, which is also the default setting for all experiments as in Sec. 4.1. We use ResNet18 as $f_p$ on CIFAR-10 dataset here. In the conventional ensemble, 30 DNNs with 5 architectures are trained with 6 seeds.

As shown in Table 4, SEP is able to halve the accuracy of AdvPoison within the same computation budget, indicating that knowledge of multiple models is much more important than additional update

steps. In comparison with the conventional ensemble, which requires $30\times$ training computation, SEP performs just a little worse. Moreover, equipped with FA, which consumes no additional calculation, efficient SEP-FA could be as effective as the conventional ensemble. With VR, SEP-FA-VR is stably better, and reduces the accuracy from 45.10% to 17.47% on average.

Table 4: Ablation study on Self-Ensemble Protection, Feature Alignment, and Variance Reduction.

| Method ($\epsilon = 2/255$) | Train (Epoch) | Crafting (Step) | CIFAR-10 Model Testing Accuracy ($\downarrow$) | | | | |
|---|---|---|---|---|---|---|---|
| | | | RN18 | SENet18 | VGG16 | DN121 | GoogLeNet |
| AdvPoison | 40 | 1000 | 41.35 | 40.54 | 52.22 | 43.28 | 48.04 |
| Ensemble | $30\times120$ | $30\times30$ | 16.32 | 16.91 | 29.19 | 16.74 | 20.34 |
| SEP | 120 | $30\times30$ | 19.92 | 17.81 | 28.12 | 18.07 | 22.48 |
| SEP-FA | 120 | $30\times30$ | 16.91 | 17.01 | 26.82 | 16.88 | 21.15 |
| SEP-FA-VR (ours) | 120 | $30\times(15+15)$ | **14.68** | **15.93** | **23.66** | **15.02** | **17.99** |

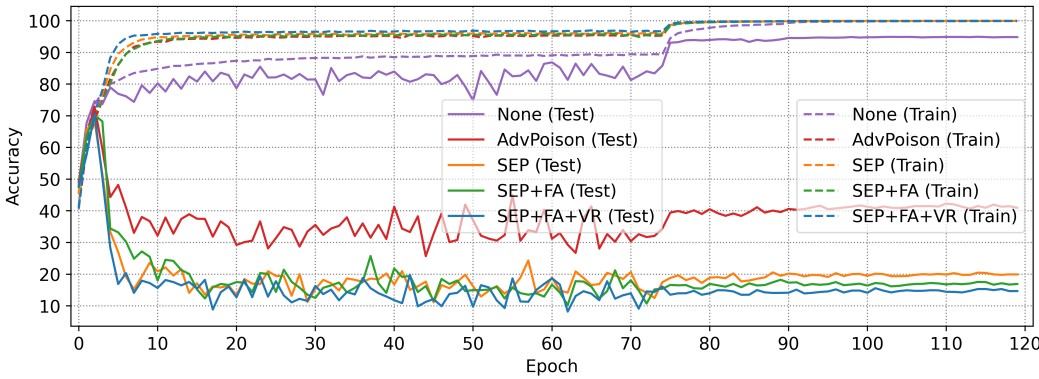

Figure 3: The accuracy trend of training a CIFAR-10 ResNet18 using different data protection methods ($\varepsilon = 2/255$, None means no protection). All methods have high training accuracy v.s. low testing accuracy, but SEP, equipped with FA and VR, reduces model's generalization most effectively.

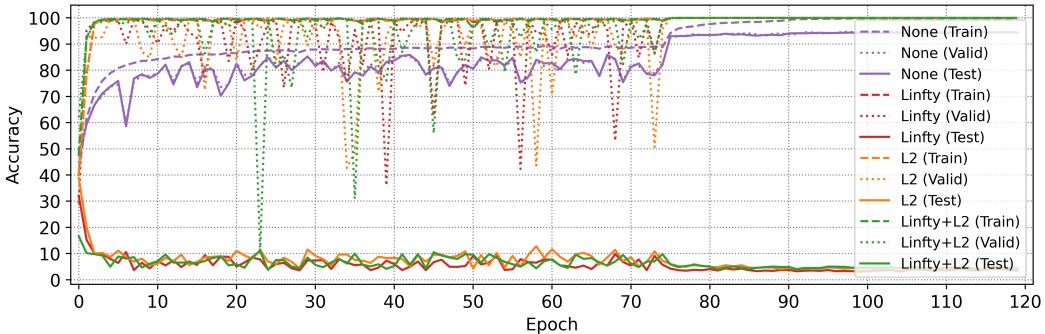

Figure 4: The validation (using 2500 samples separated from training data) performance of different types of perturbations by our method (CIFAR-10, $\ell_\infty = 8/255$, $\ell_2 = 1$, $f_p = f_a = $ ResNet18).

We illustrate the training process of Table 4 experiments in Fig. 3. Compared to the training on clean data (purple line), data protection methods accelerate the model's convergence on training data, but the DNN's testing accuracy would suddenly drop at the initial stage of training. After the learning rate decay at epoch 75, the protection performance of different methods could be clearly observed. SEP accounts for the majority of performance improvements, and FA and VR could also further decrease the accuracy, making our method finally outperform the conventional inefficient ensemble. We also show the validation accuracy (on unlearned protective examples) of different perturbations

in Fig. 4, where it is obvious that early stopping could not be a good defense because validation accuracy is mostly close to training accuracy. However, a huge and unusual gap between them may be a signal for the existence of protective examples.

We also vary the number of intermediate checkpoints $n$ used in self-ensemble to perform ablation study under different computation budgets. We set $n = 3, 5, 10, 30, 120$ without changing other hyper-parameters in self-ensemble, and plot the results in Fig. 5. Similar results could be seen, *i.e.*, FA, and VR could stably contribute to the performance. We also discover that although $n$ is increasing exponentially, the resulting performance increase would not be that significant as $n$ is large. Most prominently, raising $n$ from 30 to 120 is not bound to yield better results, meaning that the performance would saturate around $n = 30$, and it would be unnecessary to use all checkpoints.

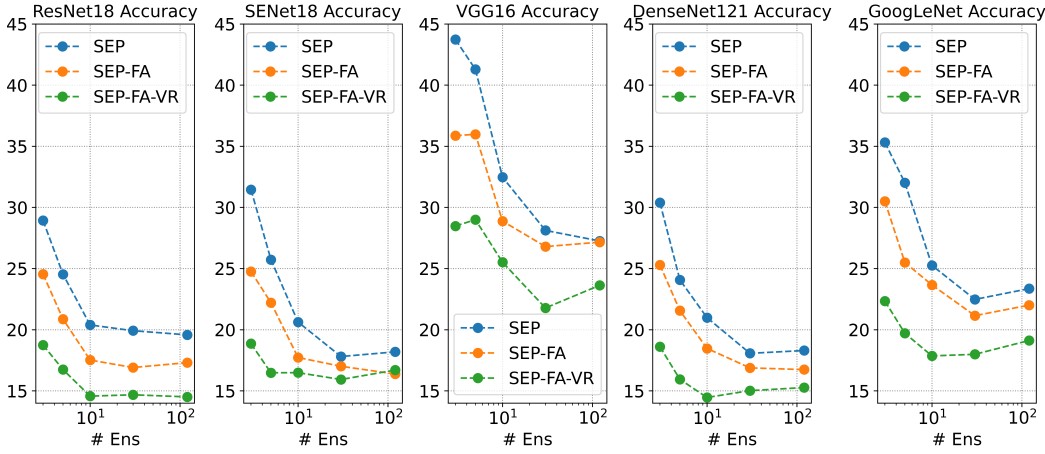

Figure 5: Ablation study on FA and VR with a different number of sub-models in self-ensemble. Results are produced on CIFAR-10 models with the bound of protective perturbations $\varepsilon = 2/255$.

## 5 DISCUSSION AND CONCLUSION

In this paper, we propose that data protection should target the vulnerability of DNN training, which we successfully depict as the examples never classified correctly in training. Such examples could be easily calculated by model checkpoints, which are found to have surprisingly diverse gradients. By self-ensemble, effective ensemble performance could be achieved by the computation of training one model, and we could also take advantage of the diverse features from checkpoints to further boost the performance by the novel feature alignment loss. Our method exceeds current baselines significantly, reducing the appropriator model's accuracy from 45.10% (best-known results) to 17.47% on average.

Our method could also serve as a potential benchmark to evaluate the DNN's learning process, *e.g.*, how to prevent DNNs from learning non-robust features (shortcuts) instead of semantic ones. And it would be interesting to study the poisoning task in self-supervised learning and stable diffusion. Since our method is implemented as the targeted ensemble attack, it is also applicable to non-classification tasks where the adversarial attack has also been developed and neural collapse also exists for pre-trained feature extractors.

## ACKNOWLEDGEMENT

This work is partly supported by the National Natural Science Foundation of China (61977046), Shanghai Science and Technology Program (22511105600), Shanghai Municipal Science and Technology Major Project (2021SHZDZX0102), and National Science Foundation (CCF-1937500). The authors are grateful to Prof. Sijia Liu for his valuable discussions.

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
