# OpenReview forum: "Self-Ensemble Protection: Training Checkpoints Are Good Data Protectors"
_ICLR.cc/2023/Conference — ICLR 2023 poster_

### Official Review · Reviewer_FTbn · 2022-10-13

**Confidence:** 4
**Correctness:** 3
**Technical Novelty And Significance:** 3
**Empirical Novelty And Significance:** 3
**Recommendation:** 6

**Clarity, Quality, Novelty And Reproducibility:**

The paper is clear enough, although the writing is hard to follow at times.  I did not notice some of the hyperparameters that are needed to re-produce the experiments, but maybe they were already there and I just did not see them.  Crucially, the hyperparameters for the competitor are important since this work does not use the same constraint space as the paper for adversarial poisons.

**Strength And Weaknesses:**

The main strength of the paper is in the empirical success compared to strong baselines and defenses.

In Figure 3, the peak test accuracy of the proposed method is just as high as for adversarial poisons.  Is there a way to reduce this peak?  Otherwise, early stopping might be a highly effective defense.

It might be worth considering the AR poisons of “Autoregressive Perturbations for Data Poisoning” as they are a recent stronger baseline than adversarial poisons.

How did you tune the attack hyperparameters (e.g. step size) of adversarial poisons since you are using a smaller perturbation radius than they used in the experiments in their paper?

In general, the writing is a bit messy and hard to follow, but the overall structure and content is good, and I was still able to understand the writing nonetheless.


**Summary Of The Paper:**

This paper proposes a simple new ensemble method for protecting data from someone training on it.  The work shows the empirical success of their method on multiple datasets and compared to strong baselines like adversarial poisons.

**Summary Of The Review:**

In general, while the writing could use some work and there is an additional baseline that would be nice to include, I currently lean towards acceptance.  The paper provides significant empirical contribution and extensive testing.

---

> ### Author Response · Authors · 2022-11-18
> **Response to Reviewer FTbn**
>
> Thanks for your insightful comments and appreciation of the clarity and experiments of our work. We address your concerns below.
>
>
> **R3.1 Comparison with an additional baseline**
>
> Thank you for mentioning this strong and newly proposed baseline. Following your suggestions, we compare with AutoRegressive (AR) poison and report the results below, which show that we outperform AR by a large margin in normal training by both $\ell_\infty$, $\ell_2$, and mixed perturbations.
>
> **Table 3: Performance of perturbations under different norms $\ell_\infty = 8 /255, \ell_2 = 1, \ell_0 = 1$ in CIFAR-10 adversarial training of ResNet18. $\ell_\infty + \ell_2$ means mixing (adding) perturbations.**
> |  Pert.  |$\ell_\infty$ | $\ell_\infty$ | $\ell_\infty$ |$\ell_2$ | $\ell_2$  |$\ell_2$| $\ell_\infty + \ell_2$ | $\ell_\infty + \ell_2$| $\ell_\infty + \ell_2$ | $\ell_\infty + \ell_0$| $\ell_\infty + \ell_0$| $\ell_\infty + \ell_0$|
> | :--: | :--: | :--: | :--: | :--: |  :--: |  :--: | :--: | :--: | :--: | :--: | :--: | :--: |
> | AT| None | $\ell_\infty$ | $\ell_2$|None | $\ell_\infty$ | $\ell_2$| None | $\ell_\infty$ | $\ell_2$|None | $\ell_\infty$ | $\ell_2$|
> |AR| 20.49| 84.73| 82.66| 12.99| 82.03| 82.86| 12.29| 82.87| 83.47| 15.25| **12.28**| **70.26**|
> |ours | **4.73** | **82.51**| **81.91**| **3.67**| **81.52**| **81.89**| **5.05**| **81.54**| **82.25**| **3.43**| 13.19| 71.01|
>
> Since AR is claimed to resist adversarial training (AT), we also conduct a thorough study on different types of perturbations and ATs as mentioned by **Reviewer bFJ9**. $\ell_2$ perturbations are crafted in the same way as AR with a step size of 0.2, and $\ell_0$ ones mean changing one pixel [1]. Results show that our method is mostly better in AT cases, and the performance under AT largely depends on the perturbation norm instead of the method. Additional results have been put into **Sec. 4.3** thanks for your constructive suggestions.
>
> [1] One-pixel shortcut: on the learning preference of deep neural networks, arXiv 2205.12141.
>
> **R3.2 How to reduce the peaked test accuracy and avoid early stopping defense**
>
> Thank you for bringing about the early stopping defense by a validation set. Note that the validation set separated from training data is also perturbed, and our additional experiments **(Fig. 4)** show that validation accuracy is close to training accuracy. Therefore, the appropriator cannot find a good point to early stop. Moreover, the model is far from convergence at the 3rd epoch when the peak accuracy emerges, so the appropriator is unlikely to early stop at this stage.
>
> But indeed it is interesting to study how to reduce the peak testing accuracy, which is a stricter requirement than reducing the final accuracy as currently studied. We find a simple strategy to keep the accuracy low is to mix (add) the $\ell_\infty$ and $\ell_2$ perturbations, which makes the accuracy always lower than 20\% under $\ell_\infty=8/255$ and $\ell_2 = 1$, seeing **Fig. 4** with additional analysis. A surprising finding here is that training with protective examples would occasionally yield a huge and unusual gap between training and validation accuracy, which may serve as a signal for shortcut detection in the future.
>
> **R3.3 Hyper-parameters of baselines**
>
> Thank you for pointing out hyper-parameters of baselines, which are indeed crucial for reproduction. Following your suggestions, we present a detailed setup of baselines in **Appendix D**, where we also add more details about the compared defenses. For advPoison, we use a step size of 1/255 as in our method for $\ell_\infty=2/255$ and $\ell_\infty=8/255$ perturbations. The smallest possible step size with lots of iterations make the optimization converge well.
>
> **R3.4 The writing is a bit messy**
>
> Following your comments, we rewrite the **introduction** to highlight our contributions. We provide additional insights on training vulnerability and relevant analysis on **Sec. 3.2 and Sec. 4.2**. We also rewrite most parts of **Sec. 3.3**, where the motivation and background of FA are presented. We appreciate more guidance on this issue and are willing to continuously improve the manuscript.

---

> ### Author Response · Authors · 2022-12-05
> **Thanks for your reviewing**
>
> Dear Reviewer FTbn,
>
> Thanks again for your insightful suggestions, which motivated us to discuss comparisons with AutoRegressive (AR) Poison and reducing the peak accuracy by the content of an additional page. We would like to also convey thanks from the authors of AR Poison (who we contacted for method details) for recognizing their work.
>
> Currently, we have addressed concerns from other reviewers, who kindly raised the scores. We are enthusiastically eager to discuss with you since we are not allowed to respond to your valuable opinions soon.
>
> Best wishes,
>
> Anonymous author(s) of Paper5517

---

### Official Review · Reviewer_bFJ9 · 2022-10-24

**Confidence:** 3
**Correctness:** 4
**Technical Novelty And Significance:** 2
**Empirical Novelty And Significance:** 2
**Recommendation:** 6

**Clarity, Quality, Novelty And Reproducibility:**

Novelty lies in using a feature alignment loss and an ensemble of snapshots instead of a single DNN to attack the images. The paper is well written and clear.

**Strength And Weaknesses:**

Strength:
1) The paper is clear and very well written.
2) Lots of experiments to support the paper claims with interesting ablation studies.

Weakness and suggestions:
1) Adversarial training easily circumvents the data corruption. The proposed method suffers from the same drawback of other existing methods.
2) L2 perturbation(not only L-inf) could complete the study. Maybe mixed perturbations could be helpful against adversarial training.
3) What about exponential moving average as a comparison to the ensemble of snapshots? It would be a way to avoid storing all these snapshots.
4) Typos in the abstract and other places for the perturbation bounds: with l_inf=8 instead of L_inf=8/255 for examples.

**Summary Of The Paper:**

This paper is about perturbing images such that it is not possible to use them for training a model which has good performance. The authors propose to build upon existing works which use adversarial perturbations to corrupt the training set. As contributions, this work proposes to ensemble the perturbation gradients coming different model snapshots of a same training deep network. The perturbation gradients come from a feature alignment loss such that perturbed samples when passed to the DNN result in feature maps which are close to the mean feature of other classes.

**Summary Of The Review:**

The paper is enjoyable to read but it only marginally improves over existing methods both technically and experimentally.

---

> ### Author Response · Authors · 2022-11-18
> **Response to Reviewer bFJ9 (2/2)**
>
> **R2.3 SEP marginally improves technically and experimentally**
>
> In our point of view, the most meaningful finding is that there are examples never classified correctly during training. Then we technically propose the notion that data protection should target at the vulnerability of DNN training. To uncover such examples, SEP is developed based on our discovery that checkpoints' gradients are surprisingly diverse. And equipped with FA that also resorts to diverse features, our method achieves effective ensemble protection by the computation of training only one DNN for the first time.
>
> Experimentally, we make possible data protection by extremely small perturbations. By setting $\ell_\infty = 2/255$, SEP perturbations on the CIFAR-10 training set reduce the testing accuracy of a CIFAR-10 ResNet18 from 94.56\% to 14.68\%, while the best-known results could only reach 41.35\%. State-of-the-art performance is also observed on CIFAR-100 and ImageNet subset. Our empirical success, as acknowledged by **Reviewer FTbn**, relies on no additional computation. This efficiency is especially important for data protection. Because only a large amount of data, if stolen, could be used to train competitive DNNs. In this regard, the scale of data requiring particular protection would be large, and saving the calculation of training extra models makes a significant difference.
>
> We rewrite the **introduction** to highlight our contributions motivated by your comments and also correct the typos thanks to your careful reading.

---

> > ### Comment · Reviewer_bFJ9 · 2022-11-29
> > **Thank you for the rebuttal**
> >
> > The authors did a great job with the rebuttal which clarified my interrogations and even if some results are limitations of the method (i.e. the adversarial training results), they provide a very useful insight for future research directions so I am upgrading my rating.

---

> > > ### Author Response · Authors · 2022-11-30
> > > **Thank you for re-evaluating our work!**
> > >
> > > We are very grateful for your efforts in re-considering our added insights and raising your scores. We sincerely hope that, as you wish, our work could inspire future directions on uncovering DNN training vulnerabilities, despite the limitations for all $\ell_\infty$ poisons on adversarial training.

---

> ### Author Response · Authors · 2022-11-18
> **Response to Reviewer bFJ9 (1/2)**
>
> Thanks for your insightful comments and appreciation of the clarity and experiments of our work. We address your concerns below.
>
> **R2.1 Mixed perturbations to resist adversarial training**
>
> Thanks for bringing about this common issue in data protection. Indeed, all the existing $\ell_\infty$ perturbations can be cancelled out by adversarial training (AT). However, it does not hinder the practicability of data protection methods because AT significantly decreases the accuracy and requires several-fold training computation. But indeed it would be interesting to study the effect of different types of perturbations in different AT settings.
>
> Following your suggestions, we study how different types of perturbations resist AT in different settings along with AR Poison [1] mentioned by **Reviewer FTbn (R3.1)**, which is claimed to resist AT. We set the perturbation $\ell_2=1$ and $\ell_0=1$, the same setting as [1] and [2], respectively. $\ell_2$ perturbations are crafted in the same way as [1] with a step size of 0.2, and $\ell_0$ ones mean changing one pixel. We keep the AT bound the same as the perturbations bound, and find that in this case, (1) both $\ell_\infty$ and $\ell_2$ ATs could recover accuracy of $\ell_\infty$, $\ell_2$, and $\ell_\infty + \ell_2$ perturbations; (2) the only type of perturbations able to resist $\ell_\infty$ AT is the mixture of $\ell_\infty$ and $\ell_0$ ones, which agrees with your guess on the effectiveness of mixed perturbations; (3) our method is always better than AR Poison in normal training by a large margin, and mostly in AT cases. Results demonstrate that the resistance to AT largely depends on the type of perturbations as you guess, instead of the data protection method. Additional results have been put into **Sec. 4.3**. Thanks for your helpful suggestions.
>
> **Table 3: Performance of perturbations under different norms $\ell_\infty = 8 /255, \ell_2 = 1, \ell_0 = 1$ in CIFAR-10 adversarial training of ResNet18. $\ell_\infty + \ell_2$ means mixing (adding) perturbations.**
> |  Pert.  |$\ell_\infty$ | $\ell_\infty$ | $\ell_\infty$ |$\ell_2$ | $\ell_2$  |$\ell_2$| $\ell_\infty + \ell_2$ | $\ell_\infty + \ell_2$| $\ell_\infty + \ell_2$ | $\ell_\infty + \ell_0$| $\ell_\infty + \ell_0$| $\ell_\infty + \ell_0$|
> | :--: | :--: | :--: | :--: | :--: |  :--: |  :--: | :--: | :--: | :--: | :--: | :--: | :--: |
> | AT| None | $\ell_\infty$ | $\ell_2$|None | $\ell_\infty$ | $\ell_2$| None | $\ell_\infty$ | $\ell_2$|None | $\ell_\infty$ | $\ell_2$|
> |AR| 20.49| 84.73| 82.66| 12.99| 82.03| 82.86| 12.29| 82.87| 83.47| 15.25| **12.28**| **70.26**|
> |ours | **4.73** | **82.51**| **81.91**| **3.67**| **81.52**| **81.89**| **5.05**| **81.54**| **82.25**| **3.43**| 13.19| 71.01|
>
> [1] Autoregressive perturbations for data poisoning, NeurIPS 2022.
>
> [2] One-pixel shortcut: on the learning preference of deep neural networks, arXiv 2205.12141.
>
> **R2.2 Can we use a moving average of snapshots to save storage**
>
> Thanks for bringing in this interesting design. Accordingly, we use the exponential moving average (EMA) of checkpoints as $m_{k+1} = \mu \cdot m_k + (1-\mu) \cdot w_{k}$, where $m$ stands for the momentum item and $w$ is for the weights. By setting $\mu$ of different values on EMA of the protector model, we always cannot craft valid protective perturbations to reduce the appropriator's accuracy, seeing the table below.
>
> **Table A: Performance of perturbations from an exponential moving average CIFAR-10 ResNet18 ($\ell_\infty = 2 / 255$)**
> |$\mu$ | 0.0 | 0.2| 0.4| 0.6| 0.8|
> | :--: | :--: | :--: | :--: | :--: |  :--: |
> |Acc (protector)| 94.52| 94.53| 94.49| 94.54| 94.44|
> |Acc (appropriator)| 93.75| 93.45| 94.37| 93.80| 93.71|
>
> Indeed, this EMA would certainly save storage. However, as shown above, it cannot effectively protect data because self-ensemble protection (SEP) targets the vulnerability of DNN training, and a single averaged DNN cannot well represent the dynamic training process. Also, SEP's success relies on the gradient diversity of checkpoints as in Fig. 1. And averaging the weights, though recording the mean of parameters, loses their variance.

---

### Official Review · Reviewer_JA31 · 2022-10-25

**Confidence:** 4
**Correctness:** 3
**Technical Novelty And Significance:** 2
**Empirical Novelty And Significance:** 3
**Recommendation:** 6

**Clarity, Quality, Novelty And Reproducibility:**

Sec 3 is not well explained, it could be rewritten for better understanding. The quality is acceptable. They have done minimal experiments to support their claims. It needs more experiments. Their self-ensemble idea is interesting, but it is mainly an extension of previous work.

**Strength And Weaknesses:**

**Strength: **

- The objective is straightforward and the problem of data protection is exciting and vital.

- Improving performance with intermediate checkpoints from a single training is an excellent idea to save time and resources.

**Weaknesses:**

- The feature alignment section could be explained a little better. Especially the neural collapse theory should be explained a little more since it is the primary tool used to develop FA.

- The experiments are limited. More results on different datasets comparing different models should have been shown. Only one model and dataset are tested while the proposed method is compared with the previous methods in Table 1.

- It is unclear what model and dataset are used to produce Table 3.


**Summary Of The Paper:**

The paper proposed to utilize intermediate checkpoints from a single training process to protect data from unauthorized use. It also proposed a novel feature alignment (FA) technique that improves the accuracy of its proposed self-ensemble protection (SEP) method. FA uses an existing theory called neural collapse to align the last layer feature of a sample for each checkpoint to the feature mean of the target class samples. It has minor improvement in performance. But the method is simple which will make it useful.

**Summary Of The Review:**

The paper attempts to solve a compelling problem and proposes a simple solution. However, it could not support empirically enough. As mentioned in the weakness section, it needs a little more work and rewriting.

***
The authors have clarified the raised doubts, updating the ratings accordingly.
***

---

> ### Author Response · Authors · 2022-11-18
> **Response to Reviewer JA31**
>
> Thanks for your insightful comments and appreciation of our motivation and idea. We address your concerns below.
>
> **R1.1 Explanation on feature alignment with neural collapse**
>
> Thanks for your careful reading. Following your advice, we further describe the background of neural collapse (NC) theory. Besides, we also describe connections between self-ensemble protection (SEP) and NC, and NC and feature alignment (FA) in **Sec. 3.3**.
>
> **SEP and NC.** Multiple checkpoints offer us a pool of features for an input. Those representations, though distinctive, all contribute to accurate classification. Thus, we could additionally take the advantage of diverse features besides diverse gradients at no cost. NC is adopted in SEP because it unravels the characteristics of DNN features very well.
>
> **Illustration of NC.** We summarize four manifestations of NC, which are (1) In-class variability of last-layer activation collapse to class means. (2) Class means converge to simplex equiangular tight frame. (3) Linear classifiers approach class means. (4) Classifier converges to choose the nearest class mean. This would provide readers with a more concrete picture of NC phenomenon.
>
> **NC and FA.** The four manifestations demonstrate that the last-layer features of well-trained DNNs would center closely on their class means. In this regard, the mean feature of in-class samples is a highly representative depiction of this class. Based on this, we develop FA that encourages the feature to approximate the mean feature of the target class. By inducing an incorrect NC on all checkpoints, FA injects non-robust features from target classes very successfully.
>
> **R1.2 Comparative experiments on more datasets and models**
>
> Thank you for helping us make our work more solid. Following your comments, we perform comparative experiments with the 3 strongest baselines on CIFAR-10/CIFAR-100/ImageNet subset and ResNet18/VGG16/DenseNet121 architectures. Results are shown in **Table 1** and comparison with weaker baselines is moved to Appendix D. Results show that we also effectively reduce the accuracy of CIFAR-100 (from 3.84\% to 2.74\% averagely) and ImageNet subset (from 4.17\% to 0.27\% averagely) models from the best-known method.
>
> **Table 1: Model testing accuracy trained on the protected dataset ($\ell_\infty=8/255$).**
> |  Dataset   |CIFAR-10 | CIFAR-10 | CIFAR-10 |CIFAR-100 | CIFAR-100  |CIFAR-100| ImageNet subset | ImageNet subset| ImageNet subset |
> | :--: | :--: | :--: | :--: | :--: |  :--: |  :--: | :--: | :--: | :--: |
> | Model| RN18 | VGG16 | DN121| RN18 | VGG16 | DN121| RN18 | VGG16 | DN121|
> |ULEs | 19.93 | 28.34 | 20.25 | 14.81 |17.56 |13.71 |12.20 |11.14 | 15.44 |
> |RULEs | 27.09 | 28.17 | 24.96 | 10.14 | 14.39 | 13.96 | 13.74 | 12.77 | 14.36 |
> |AdvPoison | 6.25 | 6.88 | 6.22 | 3.49 | 4.46 | 3.57 | 2.30 |5.40 | 4.80 |
> |**Ours**| **4.73**| **5.61**|**3.76**| **2.65** |**3.15** |**2.43** |**0.00** |**0.60** |**0.20** |
>
> We also compare with a recent baseline AutoRegressive Poison [1] mentioned by **Reviewer FTbn (R3.1)**. Results show that our method is always better in normal training by a large margin, seeing the added **Table 3**. Table 4 (the original Table 3) uses ResNet18 to craft CIFAR-10 protective examples, which we highlight in the new version.
>
> [1] Autoregressive perturbations for data poisoning, NeurIPS 2022.
>
>
> **R1.3 SEP is mainly an extension**
>
> Self-ensemble is not an extension of the conventional ensemble because they are differently motivated. Ensemble attacks aim to produce architecture-invariant adversarial examples, and such transferable examples reveal the common vulnerability of different architectures [2]. Self-ensemble protection, in contrast, targets the vulnerability of DNN training process, which is based on a surprising finding that there are samples that are ignored during the whole normal training process. SEP, enforcing consistent mis-classification, is then a novel method to find those ignored examples. Self-ensemble has never been explored in data protection due to the lack of insight on training vulnerability and study on checkpoints' diversity, and after our analysis, it would be natural to develop SEP and improve data protection in a simple and efficient way. Empirically, we reduce the accuracy in small perturbations by 60\%, which for the first time demonstrates the possibility of protecting data by extremely slight modifications. We rewrite the **introduction** to highlight our contributions motivated by your comments.
>
> [2] Universal Adversarial Attack on Attention and the Resulting Dataset DAmageNet, TPAMI 2020.

---

> > ### Author Response · Authors · 2022-12-02
> > **Thank you for re-evaluating our work!**
> >
> > We sincerely thank you for the approbation of our rebuttal. And we really appreciate the kind reconsideration of your score. Should you have any questions, please do not hesitate to discuss them with us.

---

### Author Response · Authors · 2022-11-18
**General Response**

Dear Program Chairs, Area Chairs, and Reviewers,

First of all, we would like to thank you for your time, constructive critiques, and valuable suggestions, which greatly help us improve the work. We are also grateful that reviewers unanimously regard our work as interesting and effective. Below we would like to first respond to issues concerning the **novelty and experiments** in general.

We reorganized our contributions with additional insights and explanations as suggested by **Reviewer JA31 (R1.1, R1.3), bFJ9 (R2.3), and FTbn (R3.4)**. We for the first time point out that the protective perturbations should **hurt the entire training process**. We then depict the vulnerability of DNN training, which has not been well investigated, by the perturbed training data never predicted correctly during training. This depiction is successful because such examples are always mis-classified in **any training processes**, seeing our additionally provided analysis in **Sec. 4.2**. Based on the above insights, self-ensemble is proposed as a simple way to find such examples rather than a naive extension from the conventional ensemble (in fact, no one has successfully applied regular ensemble for data protection, since it is too costly to well-train and collect many, e.g., tens of, models for ensemble attacks.)

Extra comparative experiments are also conducted following **Reviewer JA31 (R1.2), bFJ9 (R2.1), and FTbn (R3.1, R3.2)**. For normal training, besides 8 baselines on CIFAR-10 ResNet18, we also compare with 4 strongest baselines on 3 architectures on CIFAR-100/ImageNet subset **(Sec. 4.3)**. Results show that we also effectively reduce the accuracy of CIFAR-100 (from 3.84\% to 2.74\% averagely) and ImageNet subset (from 4.17\% to 0.27\% averagely) models from the best-known method **(R1.2)**. For adversarial training, we conduct a thorough study on different poisons ($\ell_\infty, \ell_2$, mixed) and their resistance to different adversarial training ($\ell_\infty, \ell_2$) compared to a new baseline **(R3.1)**. Results demonstrate that mixed perturbations could resist $\ell_\infty$ AT ($<$14\% accuracy) **(R2.1)** and reduce the peak test accuracy **(R3.2)** to below 20\%. We also discuss with **Reviewer bFJ9 (R2.2)** an interesting idea to save storage.

We carefully modified the manuscript with new insights, explanations, and experiments. All changes have been marked in yellow with the specifications of Reviewers in red. Your input contributed to a significant improvement of the paper and our proposed method as well.

Best wishes,

Anonymous author(s) of Paper5517

---

### Decision · Program_Chairs · 2023-01-20

**Decision:**

Accept: poster

**Justification For Why Not Higher Score:**

The result is quite incremental. It's a neat result that will move the field forward and appears deserving of publication, but it isn't earthshattering in a way that requires the attention of the broader community. If it did get a spotlight, it would be because this was a new problem to me (and perhaps to many others) that deserves to be highlighted. It's a cool problem that is increasingly applicable as models are trained from giant web scrapes.

**Justification For Why Not Lower Score:**

The reviewers had a consensus that the paper deserved to be accepted. They were all pleased with the novelty of the idea and the sufficiency of the results. I think this paper is generally pretty incremental, so I wouldn't be upset if this paper were to be rejected. The technical results are solid, but the novelty/distinction from previous work is the biggest point of concern.

**Metareview: Summary, Strengths And Weaknesses:**

**Summary:** This paper takes on the problem of poisoning examples so that they are hard to train on. It is a defense against attackers who  seek to collect data in an unauthorized way (e.g., via web scraping) from an entity that wishes to share data publicly (e.g., a website storing art). The defense takes advantage of two properties of neural networks: (1) neuron collapse and (2) that gradients are very different for different checkpoints throughout training. The method involves applying gradients from different checkpoints in training to each example. Doing so seems to productively prevent new networks from learning on these modified examples.

**Strengths:** This method is quite simple in practice, and it is very efficient insofar as it doesn't require an ensemble of separately trained models to produce the results. It seems like a nice advance over previous work in this area.

**Weaknesses:** The paper is a bit hard to make sense of, and the details of the algorithm aren't that clearly described in plain text. I urge the authors to take another pass over the writing to make the paper more accessible. In addition, it's unclear (at least to me) how this method would work in non-classification settings. How would this apply to something like stable diffusion or even a self-supervised image pre-training task? Would the same protection work? Is neuron collapse at play here?

**Overall:** All of the reviewers were in favor of acceptance. This is a nice incremental result, and I agree with their assessment. It is a bit more on the incremental side, so I wouldn't be heartbroken if the paper were rejected.

**Note From Pc:**

if the above contains the word "oral" or "spotlight" please see: "oral" presentation means -> notable-top-5% and "spotlight" means -> notable-top-25%. As stated in our emails, we are disassociating presentation type from AC recommendations

**Summary Of Ac-Reviewer Meeting:**

N/A